# The Impacts of Background Music on the Effects of Loving-Kindness Meditation on Positive Emotions

**DOI:** 10.3390/bs14030204

**Published:** 2024-03-04

**Authors:** Quan Tang, Jing Han, Xianglong Zeng

**Affiliations:** Beijing Key Laboratory of Applied Experimental Psychology, National Demonstration Center for Experimental Psychology Education (Beijing Normal University), Faculty of Psychology, Beijing Normal University, Beijing 100875, China; quan_tang@mail.bnu.edu.cn (Q.T.); 202128061017@mail.bnu.edu.cn (J.H.)

**Keywords:** loving-kindness meditation, background music, positive emotions, self-help interventions, compassion

## Abstract

Loving-kindness meditation (LKM) has been widely used in promoting mental health, with positive emotions as an important mechanism. The current study explored the impact of background music on the effects and difficulties of LKM practice. Two hundred participants were randomly divided into six groups, wherein LKM plus music with harmony only, LKM plus music with harmony and melody, and LKM without music were presented in a different order during the intermediate three days of a five-day LKM intervention. Participants reported three types of positive emotions (pro-social, low-arousal, and medium-arousal positive emotions) and the difficulties during meditation (lack of concentration and lack of pro-social attitudes) after each of three sessions. The results of MANOVA indicated that compared to the session without music, incorporating music could evoke more low-arousal positive emotions and pro-social positive emotions without altering the difficulties. However, the results did not reveal significant differences in the effects of music with harmony and music with harmony and melody on both emotions and difficulties. Additionally, practice effects may have influenced the generation of medium-arousal positive emotions and the difficulty of concentration, but the results were inconsistent across groups. Our findings suggest potential benefits for practitioners of LKM in incorporating music during the meditation process, and the directions for future research were further discussed.

## 1. Introduction

### 1.1. Loving-Kindness Meditation

The outbreak of the COVID-19 pandemic has had a tremendous impact on people’s mental health. A meta-analysis on the mental health of the general population during the COVID-19 pandemic indicates an average prevalence of depression at 34.31%, anxiety at 33.33%, and psychological distress at 37.54% [1]. In the current post-pandemic era, people are striving to recover their lives from the impact of the COVID-19 pandemic. Consequently, they are facing more stress from economic, work, and other life aspects than ever before, posing a threat to their mental health. An interview study by de Vroege and van den Broek [2] on mental health care professionals reported that although their mental symptoms slightly decreased after the pandemic, the reported sick leave after the pandemic was higher than during the pandemic, and absences were more frequent. They also pointed out that, despite a decrease in mental symptoms over time, the quantity or severity of mental health symptoms remained high. In such circumstances, psychological practices that help people alleviate daily stress, balance the mind and body, and regulate emotions have become essential.

One psychological practice that has been widely used in promoting mental health is loving-kindness meditation (LKM). It is a specific type of meditation derived from Buddhism. The practice involves directing one’s attention to a chosen object of well-wishing, and then, in an open and accepting manner, extending the warm feeling to others [3]. Practitioners can offer well wishes to themselves or others, including phrases such as “May you be healthy and safe” or “May you be free from suffering,” with the core focus on bringing kinder and warmer attention to oneself or others [4]. The objects of well-wishing should gradually change with practice, varying in different traditions and progressing from easy to challenging. Typically, practitioners begin with themselves, followed by loved ones, neutral individuals, adversaries, and ultimately extending well wishes to all sentient beings [5]. 

An important function of LKM is to generate positive emotions. A meta-analysis has shown that loving-kindness meditation can induce immediate positive emotions, and multi-week interventions of LKM are effective in enhancing daily positive emotions [5]. Moreover, previous research has found that the elevation of positive emotions mediates other positive outcomes of LKM (e.g., increase of mindfulness, purpose in life, social support, and decrease of illness symptoms), supporting the significant role of positive emotions in LKM [3]. The emotions evoked by LKM for practitioners should be directed towards specific objects with specific levels of arousal. Hutcherson et al. [6] found that LKM can increase social connection and generate positive social emotions towards other people. Seppala et al. [7] also demonstrated that a 10 min LKM session can induce pro-social emotions such as friendliness, closeness to others, affection, and love. Additionally, a study found that about 6 min LKM interventions increase low-arousal positive emotions instead of high-arousal positive emotions, and other-focused positive emotions instead of self-focused positive emotions [8]. There is additional evidence that LKM interventions tend to increase un-activated pleasant rather than activated pleasant emotions [9]. 

The relationship between LKM and positive emotions is not unidirectional; instead, positive emotions can also facilitate the practice of meditation. For instance, a study indicates that participants with higher positive emotions before accepting LKM intervention tend to have more positive emotions after the intervention, and they are also more willing to continue the meditation practice [10]. Meta-analysis from Zeng et al. [5] also reported that individual differences and the nature of positive emotions can influence the effects of LKM in generating positive emotions. What is more, a study conducted by Van Cappellen et al. [11] found that the level of positive emotions experienced during the initial meditation can predict the frequency and duration of meditation practice in the following 21 days. The more positive emotions generated during the initial loving-kindness meditation, the higher the frequency of subsequent participation in LKM practice. Moreover, when participants prioritize the positive impact of emotions, it can amplify the effect of LKM on positive emotions compared to that of mindfulness meditation.

With the spread of meditation practices, researchers and practitioners are increasingly focusing on how to enhance the effectiveness of meditation [12], and the impact of background music on meditation has received more attention. In fact, several studies have looked at the effects of background music on mindfulness meditation [12,13,14,15,16]. The effect of music on emotions is obvious, e.g., [17,18,19], and one function of LKM is to generate positive emotions [11]. Therefore, the aim of this study is to examine the effect of adding background music on LKM.

### 1.2. Impact of Background Music

Music is present in almost all cultures and serves as a significant medium for human emotional perception and expression. Human perception of music relies primarily on three structural components of music, namely melody, harmony, and rhythm [20]. Based on perceived information, the human brain processes music into emotions through three mechanisms [20]: (1) hardwired responses, referring to music eliciting universal survival-related responses, such as the brainstem triggering fear responses to loud sounds; (2) extramusical associations, indicating the connection between music and extramusical elements carrying specific emotions, such as emotional contagion, visual imagery, and episodic memories; and (3) anticipation, referring to whether the structure of music meets or disappoints the listener’s expectations. 

Numerous behavioral studies have also demonstrated the role of music in eliciting emotions, e.g., [17,18,19], especially slow music that brings low-arousal emotions, and the major mode that is associated with positive emotions [21]. Notably, research from Yu et al. [22] suggests that listening to pro-social music can lead individuals to engage in more pro-social emotions compared with neutral music because of emotional contagion which allows music to activate the emotional representations of the brain that correspond to the music, thereby eliciting similar emotions [23].

In addition to eliciting emotions, listening to music also serves other emotional regulation functions [24]. Moore [25] provided evidence in his systematic review regarding the neural correlations of music and emotion regulation, implicating the amygdala, orbitofrontal cortex, lateral prefrontal cortex, dorsolateral prefrontal cortex, and ventrolateral prefrontal cortex. These findings suggest that listening to music contributes to emotion regulation, especially when the music is perceived as joyful and pleasant or possesses predictable, consonant harmonies. Moreover, studies utilizing music as a therapeutic intervention have already demonstrated the benefits of music-related activities on individuals [26,27,28], especially for positive emotions [27,29] and pro-social attitudes [30,31]. 

In both practice and research of meditation, it is common to incorporate background music during the practice. Liu and Rice [32] found that skilled meditation practitioners prefer meditation without background music, while music without a specific melody ranks second. Beginners with no prior experience in meditation tend to prefer music without a specific melody. Hernandez-Ruiz and her colleagues conducted a series of studies examining the effect of background music on mindfulness meditation from the perspective of the stimulating characteristics of music [12,13,14,15,16]. They discussed the effects of four different musical stimuli—script (script without music), beat (script with the steady tone only), harmony (script with the tone and simple harmony), and melody (script with music)—on the effectiveness of mindfulness meditation interventions (enhancing mindfulness and reducing stress) and preferences, both for non-musicians and musicians. However, after controlling for *Absorption in Music* (referring to the emotional experience of an individual immersed in music), there were no significant differences in effects among the stimuli, and different musical stimuli did not provide additional effects on enhancing mindfulness and relieving stress. Without the control of music absorption, script with beats had a better effect on mindfulness, but participants (both non-musicians and musicians) ranked harmony and melody as the most useful and desirable musical stimuli.

To our knowledge, only one study has discussed the effects of adding background music to LKM. Sorensen et al. [33] combined classical guitar music with LKM for a 3-week intervention, and the results showed that although the intervention increased well-being, mindfulness, self-compassion, and reduced fear of compassion and self-compassion, LKM with music had no additional effect when compared with the LKM-only group and the music-only group. In the four weeks after the intervention, participants were allowed to practice LKM voluntarily, and the results showed that the total amount of practice had an effect on improving self-compassion and mindfulness, but it had no effect on reducing fear of compassion, self-compassion and improving well-being. Similarly, the extra effect of adding music to the LKM was not found. We suppose that these results may be due to the timing of the test and the choice of music. Given that the inclusion of music in LKM is novel, the detection of the effects of single-session practice may be more sensitive. In addition, according to the results of Hernandez-Ruiz and Dvorak [14], researchers may need to be more careful about the choice of music, because structures that are too complex may affect the effect of music. However, we agree with Sorensen et al. [33] about the benefits of adding music to LKM, and based on the overlapping functions of music and LKM on positive emotions, as well as the effects of music on other meditations mentioned above, we suppose that adding background music to the practice of LKM is likely to contribute to the generation of positive emotions.

One issue that should be considered when incorporating music into LKM is whether practitioners may be influenced by music, thereby not being able to focus on the practice of meditation. The existence of this concern is because listening to music is likely to take up limited attentional resources that could be used for LKM [34]. However, according to the arousal-mood theory [35,36], the level of arousal of practitioners can be affected by background music, and the positive emotions triggered by music may help them to concentrate. In fact, studies have shown that the presence of background music contributes to attentional performance, both for musicians and non-musicians [37]. Moreover, some studies have found that slow music helps concentration more effectively than fast music [38]. Research on patients with schizophrenia has also shown that adding background music, especially slow background music, can improve work attention performance [39]. Additionally, it has also been noted that music with lyrics may not be suitable for LKM because it may affect attention performance [40]. Therefore, we believe that the addition of soothing, lyrics-free music to LKM would not make it difficult for practitioners to meditate and could even improve their concentration during practice. Based on the fact that melody is generally more distinctive than harmony [41], we suppose it may be more likely that melody interferes with attention.

### 1.3. Overview

In the current study, we intended to examine the impacts of background music on positive emotion generation and difficulty of LKM. We conducted a 5-day LKM intervention and evaluated the impact of music addition during intermediate three days. Specifically, we compared LKM plus music with harmony, LKM plus music with harmony and melody, and LKM without music, and focused on different types of positive emotions (i.e., low-arousal positive emotion, medium-arousal positive emotion, and pro-social positive emotion) and difficulties during meditation (i.e., lack of concentration, lack of pro-social attitudes). The following hypotheses were proposed: 

**Hypothesis 1.** 
*Compared to LKM without music, LKM with music generates more positive emotions. In particular, LKM plus music with harmony and melody is able to generate more positive emotions than LKM plus music with harmony only.*


**Hypothesis 2.** 
*Compared to that of LKM without music, LKM with music has a lower level of lack of concentration. In particular, it is more difficult to stay concentrated in LKM plus music with harmony and melody than in LKM plus music with harmony only.*


## 2. Methods

### 2.1. Design and Participants

There were three conditions in this study: LKM plus music with harmony (condition A), LKM plus music with harmony and melody (condition B), and LKM without music (condition C). Based on the Latin square design, the three conditions were used to form six groups (ABC/ACB/BAC/BCA/CAB/CBA), and the 6 groups × 3 conditions were analyzed by MANOVA, using IBM SPSS Statistics 26. The four types of emotion and the two dimensions of difficulties during meditation were separately used as dependent variables. G*Power Version 3.1.9.2 [42] was used to estimate the sample size. Because our core objective was to detect the main effect of conditions, with Cohen’s *f* = 0.25, statistical significance *α* = 0.05, using the recommended algorithm and not using the mean correlation, 162 participants were needed to guarantee Power of 0.80.

All participants were recruited through online social networking platforms in China, including WeChat Moments, Weibo, and Xiaohongshu. A recruitment form for this study was included in a meditation course where participants could voluntarily choose whether to participate in the study. There were two recruitment rounds, each with identical recruitment and experimental procedures. At the end of the recruitment process, participants were screened according to the following conditions: (1) no music-related background (no professional learning and practice experience in music or musical instruments, non-music industry practitioners); (2) not at the onset of any mental illness or psychological disorder and no corresponding medical history. Finally, we recruited 393 participants and randomly assigned them to one of six groups. Two hundred of them completed all the experimental procedures and were included in the statistical analysis (including 187 female, *M*_Age_ = 31.68, *SD* = 6.14, ranging from 19 to 50 years old): 32 of them were in group ABC, 35 in group ACB, 34 in group BAC, 36 in group BCA, 32 in group CAB, and 31 in group CBA.

### 2.2. Measurements

*Emotions.* The Emotional Words List, originally developed by Lee et al. [43] and later adapted by Zeng et al. [44], was used to measure the levels of four categories of emotions experienced by participants during the meditation. The scale is a 9-point Likert-type self-reporting scale (ranging from 1 = *not at all* to 9 = *extremely strong*) with 4 dimensions based on 12 emotional words: medium-arousal positive emotions (delighted, happy, satisfied), low-arousal positive emotions (calm, peaceful, serene), pro-social positive emotions (love, care, friendly), and negative emotions (sad, gloomy, blue; used as fillers). Participants reported the intensity of each emotion they experienced during the meditation, with higher scores indicating stronger emotions. In the current study, the Cronbach’s *α* coefficients of the four dimensions were 0.89, 0.92, 0.86, and 0.92, respectively.

*Difficulties During Meditation.* The Difficulties during Meditation Involving Immeasurable Attitudes Scale developed by Zeng et al. [8] was used to measure the LKM practice difficulty experienced by participants. The scale is a 6-point Likert-type self-reporting scale (ranging from 1 = *not at all* to 6 = *totally agree*) with 12 items in two dimensions, namely lack of concentration (e.g., “I am constantly interfered by all kinds of irrelevant thoughts.”) and lack of pro-social attitudes (e.g., “I am cold or indifferent toward my target in practice.”). Higher scores that participants rated on this scale meant that they experienced higher meditation practice difficulty. In the current study, the Cronbach’s *α* coefficients of lack of concentration and lack of pro-social attitudes were 0.92 and 0.85, respectively. 

*Demographic Information.* We also collected demographic information of the participants, including gender, age, occupation, resident area, meditation experience, and work situation.

### 2.3. Materials

The LKM course used in this study is a 5-day program excerpted from an online self-help 21-day LKM audio course in a previous study by Zeng et al. [45], with the authorization of the researcher. Each session of the course covers a brief introduction about meditation techniques and an approximately 5 min meditation practice. Among the 5 days, the breathing technique, as a basic skill of meditation, is taught on Day 1 with no music accompanied. The technique taught over the next few days is to direct well wishes towards a chosen target with goodwill. The instructions of well-wishing are “May you be healthy and safe” and “May you be happy and joyful”. From Day 2 to Day 4, the LKM course audio requires friends of the participant to be the target of well-wishing, during which LKM plus music with harmony, LKM plus music with harmony and melody, and LKM without music were present in different order for each group of participants. For Day 5, a person for whom one feels gratitude is chosen as the target with no music added during LKM practice. 

The two background music pieces were selected from YouTube and retrieved using the keywords “peaceful meditation music” and “love gratitude meditation music”. Music A was from the YouTube channel Soothing Relaxation, with permission obtained from the creator. Music B was sourced from the YouTube channel Miracle Tones Meditation, allowing for non-commercial use. Music A was composed of harmony, while music B was composed of melody and harmony, both of which were relatively soothing as a whole. Considering that the audio was added to assist LKM, we referred to the guidance of Moore [25], and the harmony of the two selected audio was harmonious. The melody part of music B was a slowly upward arpeggio that appeared repeatedly for many times, adding positive emotional color. At the same time, referring to the research of Dvorak and Hernandez-Ruiz [12], we chose the music with harmony only and the music with harmony and melody as the materials, rather than the simple comparison of harmony and melody. We also believe that melodies that do not contain harmony at all are abrupt and not suitable for LKM. Music A and Music B were used as the background music for the 5 min LKM practice under Condition A and Condition B of Days 2–4, respectively.

### 2.4. Procedure

This study was conducted online. One week before the start of the experiment, all participants who were successfully enrolled and eligible were randomly divided into six experimental groups. All participants needed to complete a 5-day meditation course and 3 questionnaires. The meditation course was released on the Chinese online knowledge service platform Xiaoe-Tech. Prior to the start of the study, instructional manuals and exclusive course links were sent to participants via WeChat. In order to avoid the situation where the participants were not able to listen to the course every day due to their personal reasons, we extended the opening time of the course, requiring participants to complete 5 days of the course within 8 days. In this 5-day meditation program, our experiment collected practice data and measurements for the middle three days. Participants in each group then received two different background-music and no-music conditions of LKM practice in a different order on days 2–4 as noted above. They were asked to fill out a measurement questionnaire on the day after completing the sessions on days 2–4, measuring the positive emotions generated during the practice and the meditation difficulty of the practice on that day. Researchers ensured that participants could only complete one session per day by checking the backend records of the Xiaoe-Tech platform. At the end of the session, participants who completed all 3 sessions were rewarded with a meditation practice course. All participants completed informed consent at the time of recruitment, and all procedures were approved by the Ethics Committee of the Faculty of Psychology, Beijing Normal University with an IRB Number: BNU202312070200.

## 3. Results

### 3.1. Preliminary Analysis

After excluding participants who failed to fully complete all three days of measurements, we calculated the mean values and standard deviations of low-arousal positive emotion, medium-arousal positive emotion, pro-social positive emotion, negative emotion, lack of concentration, and lack of pro-social attitudes reported by participants after each time practicing, namely LKM plus music with harmony (i.e., Condition A), LKM plus music with harmony and melody (i.e., Condition B), and LKM without music (i.e., Condition C; see Table 1).

### 3.2. Hypothesis Testing

To test our hypothesis, six 6 groups × 3 conditions MANOVAs were used to examine the effect of music on a single practice of LKM (see Table 2). Except for the significant Mauchly’s test of sphericity for negative emotion (W = 0.958, *p* = 0.016), all core dependent variables met the hypothesis of sphericity (Ws ∈ [0.976, 0.997], *ps.* > 0.050). Therefore, the within-subject effects test for negative emotion was adjusted using the Greenhouse–Geisser method.

The MANOVAs found that the main effects of groups on all dependent variables were not significant for low-arousal positive emotion, *F*(5, 194) = 0.646, *p* = 0.665, *η*_p_^2^= 0.016; for medium-arousal positive emotion, *F*(5, 194) = 0.658, *p* = 0.655, *η*_p_^2^= 0.017; for pro-social positive emotion, *F*(5, 194) = 1.142, *p* = 0.340, *η*_p_^2^ = 0.029; for negative emotion, *F*(5, 194) = 1.129, *p* = 0.346, *η*_p_^2^ = 0.028; for lack of concentration, *F*(5, 194) = 1.635, *p* = 0.152, *η*_p_^2^ = 0.040; and for lack of pro-social attitudes, *F*(5, 194) = 0.132, *p* = 0.985, *η*_p_^2^ = 0.003. These results indicate that regardless of the order in which the participants received the three conditions, there were no significant differences reported in their emotions and meditation difficulties.

The main effects of conditions were significant for low-arousal positive emotion (*F*(2, 388) = 3.438, *p* = 0.033, *η*_p_^2^ = 0.017) and pro-social positive emotion (*F*(2, 388) = 3.061, *p* = 0.048, *η*_p_^2^ = 0.016). Further post hoc analysis using the widely applied least significant difference method (LSD) [46] for pairwise comparison revealed a significant difference in low-arousal positive emotion between participants who underwent LKM plus music with harmony (*t* = 0.257, *p* = 0.025, Cohen’s d = 0.148) and LKM plus music with harmony and melody (*t* = 0.253, *p* = 0.017, Cohen’s d = 0.143) compared to that of LKM without music, whereas no significant difference was observed between low-arousal positive emotion induced by LKM plus music with harmony and LKM plus music with harmony and melody (*t* = 0.004, *p* = 0.972, Cohen’s d = 0.001).

Similar to those for low-arousal positive emotion, the results for pro-social positive emotion showed significant differences between LKM plus music with harmony only (*t* = 0.225, *p* = 0.033, Cohen’s d = 0.130) and LKM plus music with harmony and melody (*t* = 0.227, *p* = 0.022, Cohen’s d = 0.130) compared to LKM without music, while no significant difference was observed between LKM plus music with harmony only and LKM plus music with harmony and melody (*t* = −0.003, *p* = 0.981, Cohen’s d = 0.007). These results indicate that compared to LKM without music, incorporating music into LKM led to greater low-arousal positive emotion and pro-social positive emotion. However, adding two different pieces of background music did not result in significant differences in the two positive emotions induced by LKM. 

What is more, for medium-arousal positive emotion (*F*(2, 388) = 0.952, *p* = 0.387, *η*_p_^2^ = 0.005), negative emotion (*F*(2, 388) = 1.372, *p* = 0.255, *η*_p_^2^ = 0.007), lack of concentration (*F*(2, 388) = 0.826, *p* = 0.438, *η*_p_^2^ = 0.004), and lack of pro-social attitudes (*F*(2, 388) = 1.167, *p* = 0.312, *η*_p_^2^ = 0.006), the main effects of conditions were not significant, which indicates that after experiencing different conditions, the participants did not show significant differences in the mentioned variables.

We found significant interaction effects between groups and conditions for medium-arousal positive emotion (*F*(10, 388) = 3.786, *p* < 0.001, *η*_p_^2^ = 0.089) and lack of concentration (*F*(10, 388) = 2.175, *p* = 0.019, *η*_p_^2^ = 0.053). Additionally, there was a marginally significant interaction between the two for pro-social positive emotion, *F*(10, 388) = 1.808, *p* = 0.058, *η*_p_^2^ = 0.045. We further analyzed this interaction too. Simple effect analyses for medium-arousal positive emotion, lack of concentration, and pro-social positive emotion were conducted using the LSD method for pairwise comparisons. 

The analysis of medium-arousal positive emotion revealed that participants in the ABC group (i.e., the order of the three conditions was music with harmony, music with harmony and melody, and no music) reported significantly more medium-arousal positive emotion after practicing LKM plus music with harmony and melody (*t* = 0.969, *p* = 0.001, Cohen’s d = 0.489) and LKM without music (*t* = 0.729, *p* = 0.008, Cohen’s d = 0.370) compared to that after LKM plus music with harmony only. However, participants in the BCA group (i.e., the order of the three conditions was music with harmony and melody, no music, and music with harmony) reported significantly more medium-arousal positive emotion after practicing LKM plus music with harmony only compared to that after LKM without music, with *t* = 0.537, *p* = 0.036, Cohen’s d = 0.288. Meanwhile, participants in the CBA group (i.e., the order of three conditions was no music, music with harmony and melody, and music with harmony) reported significantly more medium-arousal positive emotion after meditating in LKM plus music with harmony (*t* = 1.097, *p* < 0.001, Cohen’s d = 0.595) and LKM plus music with harmony and melody (*t* = 0.602, *p* = 0.031, Cohen’s d = 0.316) compared to that after LKM without music. No significant differences were observed in the remaining tests (*ps.* ∈ [0.060, 0.970], ns.). We believe that the seemingly contradictory results could be attributed to the practice effect, meaning that practices presented later were more likely to elicit more medium-arousal positive emotions. 

In the analysis of pro-social positive emotion, we found more pro-social positive emotions reported by participants in the BCA group (*t* = 0.556, *p* = 0.025, Cohen’s d = 0.338) and CBA group (*t* = 0.763, *p* = 0.004, Cohen’s d = 0.458) after practicing LKM plus music with harmony compared to that after LKM without music. Additionally, in the BCA group, participants reported stronger pro-social positive emotion after experiencing LKM plus music with harmony and melody compared to that after LKM without music, *t* = 0.630, *p* = 0.007, Cohen’s d = 0.404. The remaining results were not significant (*ps.* ∈ [0.077, 0.969], ns.). These results indicate that LKM with music can bring more pro-social positive emotion compared to LKM without music, although it only appeared in separate individual groups.

A simple effect analysis for lack of concentration revealed that for participants in the ACB group (i.e., the order of the three conditions was music with harmony, no music, and music with harmony and melody), LKM plus music with harmony was more effective in focusing on LKM compared to LKM without music, *t* = −0.424, *p* = 0.016, Cohen’s d = 0.479; for participants in the BCA group, LKM plus music with harmony and melody was more effective in concentration compared to LKM without music, *t* = −0.384, *p* = 0.022, Cohen’s d = 0.324; and for participants in the CBA group, it was easier to stay on track with LKM when practicing LKM plus music with harmony compared to LKM plus music with harmony and melody, *t* = −0.398, *p* = 0.034, Cohen’s d = 0.331. The remaining analyses for participants’ reports on lack of concentration were not significant (*ps.* ∈ [0.072, 0.826], ns.). In separate individual groups, we observed that maintaining concentration was easier under the condition of LKM with music compared to the condition of LKM without music.

Finally, we did not find significant interaction effects between groups and conditions in low-arousal positive emotion (*F*(10, 388) = 1.017, *p* = 0.428, *η*_p_^2^ = 0.026), negative emotion (*F*(10, 388) = 1.373, *p* = 0.194, *η*_p_^2^ = 0.034), and lack of pro-social attitudes (*F*(10, 388) = 1.250, *p* = 0.257, *η*_p_^2^ = 0.031), which indicates that participants practicing in different sequences did not show significant differences in their experiences of the mentioned variables under different conditions.

## 4. Discussion

The current study evaluated the impact of background music on LKM. We detected a facilitative effect of adding music on the effectiveness of LKM in eliciting positive emotions. Compared to the no-music condition, participants reported more low-arousal positive emotions and pro-social positive emotions after engaging in LKM with music. These results provide supportive evidence for the first time that music enhances the effectiveness of LKM practice, aligning with previous research on music’s impact on emotions, e.g., [47], and the effects of incorporating music into other meditation practices, e.g., [48]. As for the inconsistency between the results of this study and those of previous research on LKM with music [33], it may be attributed to this study focusing on immediate effects rather than daily positive emotions, which implies that the impact of adding music on enhancing daily positive emotions or intervening in long-term effects is still worth investigating. It is worth noting that this study cannot directly reveal whether the increase in positive emotions is due to music enhancing the LKM practice or simply because music itself induces positive emotions. However, regardless of the mechanism, the inclusion of music undeniably enhances the intended effects of LKM.

Notably, this study found that the addition of background music had a promoting effect only on the generation of low-arousal and pro-social positive emotions without affecting medium-arousal positive emotions, which is reasonable. What emotions music can induce may depend on the valence and arousal conveyed by the music itself. Research indicates that cognitive benefits related to music exposure only occur when the emotional expression of the music can match the participant’s emotional state (both valence and arousal level) [49]. In our study, medium-arousal positive emotions increased with the number of practice sessions but were not influenced by background music. We believe this may have been because the two pieces of music we chose could only induce a lower level of arousal, unable to evoke the corresponding medium-arousal positive emotions. Evidence from electroencephalography (EEG) studies also suggests that slow-paced music induces lower emotional arousal, supporting the above explanation [50]. Additionally, emotional contagion may be a reason for the difference in the generation of pro-social positive emotions. A review by Juslin and Västfjäll [23] points out that the emotions expressed in music are mimicked internally by the listener, inducing the same emotions. Due to considerations of meditation quality, we have chosen music that is more suitable for LKM, allowing the addition of music to bring more pro-social positive emotions.

When considering the disparities brought about by music with different stimuli, we did not observe different effects of background music containing only harmony or containing both harmony and melody on the positive emotions generated by LKM. For low-arousal and pro-social positive emotions, the difference between the two types of music seemed to be nonexistent. Regarding medium-arousal positive emotions, there was a significant interaction between the practice order and music condition. Such results may indicate that the effects of the presence or absence of melody were not stable, and the impact of background music with melody on medium-arousal positive emotions could only be detected under specific practice order conditions. We suppose that this may have been, as pointed out by Franco et al. [49], because both types of music had relatively low-arousal levels and positive valence. Emotional contagion [23] can also support this interpretation, that is, because of the high degree of similarity (both arousal and pro-sociality) of the two pieces of music, the emotions the listeners mimicked and generated were similar. Therefore, the differences in their effects on emotion induction, corresponding to arousal and valence, could not be clearly distinguished. Additionally, considering a meta-analysis [5] that suggests that positive emotions increase continuously with the practice of LKM, we believe that significant results in individual groups occurred because the effects of melody were only found when the practice order and effect direction were consistent. Future researchers may consider using more distinctive audio materials or larger sample sizes to detect the effects.

In terms of the effect of background music on the difficulty of the meditation, our hypothesis was not completely validated. The main effect of adding soothing, lyric-free music did not make LKM more difficult or easier, although we separately found that participants in individual groups were more likely to focus attention when they practiced LKM with music rather than LKM without music, as well as more likely to focus attention when they practiced LKM without melody instead of LKM with melody. These results align with the findings of Hernandez-Ruiz et al. [16] in mindfulness meditation, although they did not find a greater impact of adding music on stress reduction compared to single mindfulness meditation sessions. However, the stress-reducing effect of mindfulness meditation with added music remained significant, indicating that the addition of music did not interfere with the meditation process. Additionally, they did not find differential effects of the presence or absence of music on negative emotions, which is highly consistent with this study. We believe this is because music creates an ideal environment for meditation, carrying suitable arousal and valence. According to the arousal-mood theory [35,36], auditory stimuli can enhance cognitive performance, particularly by adjusting physiological arousal and improving mood. The inclusion of background music in LKM can regulate the practitioner’s arousal to an appropriate level during the process, and the positive emotions induced by music counteract the psychological resource consumption caused by listening to music, ensuring the practitioner’s attention to meditation. The result of increased attention due to background music has been replicated in multiple studies [37,38,39,51]. Furthermore, we found that background music with melody made it more difficult to concentrate than music with harmony did only in some individual groups. The reason we could find this may be similar to that with positive emotions, namely, the addition of melody occupies attentional resources, but this effect is weak and can only be observed when it aligns with the direction of the practice effect. In summary, this study did not find strong evidence that music enhances or interferes with concentration during LKM.

This study also has the following limitations. First, for the sake of meditation effectiveness and participant interest, the music selected in this study, although inherently different (whether or not it included melody), all belonged to the more soothing category. Future researchers may consider examining the effects of music of different styles, tempos, or bringing different subjective experiences on LKM. This will help practitioners determine or create the best supportive conditions, even including specific pieces of music. Second, this study focused on the positive emotions brought about by single-session meditation, but we still know little about daily emotions and emotion-related psychological functions (such as emotion regulation, emotion recognition, emotion expression). Future research could consider discussing the effects of long-term LKM interventions with music on practitioners. Third, the mechanism of music affecting LKM is not yet clear. Although we have found positive effects of adding music to LKM with practical significance, considering that we detected the effect of adding music on concentration in some participant groups, we speculate that the impact of music on LKM may not be a simple parallel addition. Therefore, it is still necessary to discuss potential mediating variables or regulatory mechanisms. In addition, researchers may consider neural or physiological indicators to more visibly reveal the mechanisms of music’s effects. Finally, none of the participants screened in this study had experience with LKM. Given the practice effect mentioned above, future research could consider providing background music to experienced LKM practitioners, which may allow the role of music to be highlighted. 

## 5. Conclusions

Our findings suggest that the addition of simple-structured, soothing, lyrics-free music helps enhance the positive emotions generated by LKM and does not make the practice more difficult. In future LKM practice, appropriate music may be incorporated in the process for its full potential effect on obtaining a better outcome.

## Figures and Tables

**Table 1 behavsci-14-00204-t001:** Descriptive statistical results (*n* = 200).

	LKM Plus Music with Harmony		LKM Plus Music with Harmony and Melody		LKM Plus without Music	
	*M*	*SD*	*M*	*SD*	*M*	*SD*
LPE	6.59	1.66	6.59	1.79	6.34	1.71
MPE	5.37	1.82	5.47	1.92	5.33	1.83
PPE	6.31	1.65	6.32	1.86	6.09	1.76
NE	2.16	1.60	2.30	1.71	2.13	1.58
LOC	2.98	1.16	3.03	1.12	3.09	1.11
LOP	2.22	0.93	2.14	0.87	2.24	0.95

Note. LPE = low-arousal positive emotion, MPE = medium-arousal positive emotion, PPE = pro-social positive emotion, NE = negative emotion, LOC = lack of concentration, LOP = lack of pro-social attitudes.

**Table 2 behavsci-14-00204-t002:** Results of MANOVA (*n* = 200).

	Main Effect of Group *df* (5, 194)	Main Effect of Condition *df* (2, 388)	Interaction of Group and Condition *df* (10, 388)
LPE	0.646 (0.665, 0.016)	3.438 (0.033, 0.017)	1.017 (0.428, 0.026)
MPE	0.658 (0.655, 0.017)	0.952 (0.387, 0.005)	3.786 (<0.001, 0.089)
PPE	1.142 (0.340, 0.029)	3.061 (0.048, 0.016)	1.808 (0.058, 0.045)
NE	1.129 (0.346, 0.028)	1.372 (0.255, 0.007)	1.373 (0.194, 0.034)
LOC	1.635 (0.152, 0.040)	0.826 (0.438, 0.004)	2.175 (0.019, 0.053)
LOP	0.132 (0.985, 0.003)	1.167 (0.312, 0.006)	1.250 (0.257, 0.031)

Note. LPE = low-arousal positive emotion, MPE = medium-arousal positive emotion, PPE = pro-social positive emotion, NE = negative emotion, LOC = lack of concentration, LOP = lack of pro-social attitudes. Each of these cells is *F* (*p*, *η*_p_^2^).

## Data Availability

The raw data supporting the conclusions of this article will be made available by the authors upon request, without undue reservation.

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
