# Peer review of "The Impacts of Background Music on the Effects of Loving-Kindness Meditation on Positive Emotions"

_behavsci, 2024, doi:10.3390/bs14030204_

Round 1
Reviewer 1 Report
Comments and Suggestions for Authors
I am not very convince with the music with harmony or melody concept of intervention. These notions are very abstract . I do not know how authors can objectify the same.
Author Response
We would like to express our sincere gratitude to you. Thank you for the time and effort you put into reviewing this article, as well as your suggestions for the manuscript. We have responded to your comments point by point, please see the attachment.

Reviewer 2 Report
Comments and Suggestions for Authors
A very interesting paper. One point on English language - the authors refer to the practice of 'Blessings', I wonder whether calling it 'Well wishing" might be better understood.
Did researchers ask whether people liked the different pieces of music or not? If so, did this have an impact on their results?
Author Response
We would like to extend our gratitude for the time and effort you have put into this manuscript and for the valuable suggestions you have given us. We have responded to your comments point by point, please see the attachment.

Reviewer 3 Report
Comments and Suggestions for Authors
The topic sounds interesting but the paper is not well organized and the study has major defect. There are some point need further explain, For example, about the rationales for this study. The organization of the introduction section still needs improvement. By reading the other parts of the manuscript, it seems that the test of mediations was the primary purpose, and comparing the mediational relationships.
The section on methodology requires further explanation, as well as clarification of statistical procedures. Especially the differences between groups.
Overall, it is very confused about the rationales for this study. In addition, need more explanation in the discussion section and need to provide information about research gaps and future perspectives.
Comments on the Quality of English LanguageAs for the English language, it needs a slight modification regarding scientific writing.
Author Response
Thank you for the time and effort you put into reviewing our manuscript. Your comments and suggestions have helped us effectively improve the quality of our articles. We have responded to your comments point by point, please see the attachment.

Reviewer 4 Report
Comments and Suggestions for Authors
The study is timely for research on compassionate mindfulness practice. It is recommended to specify the following points.
In the introduction, the studies mentioned are useful to justify the use of this type of LKM meditation. However, there is a missing link between the first part of the introduction and the second (Impact of Background Music). It is advisable to end a section where the subsequent one is introduced.
Method Procedure The location and conditions under which the LKM was practiced (and its specific instructions, its narration), nor the use of the platform is unclear. It is convenient to illustrate step by step, to replicate it.
Results
The results are presented in very long paragraphs. They should be limited to between 4 to 6 lines, as it makes them difficult to read. It is possible to include results broken down by subsections or develop them by table. In multiple comparisons, comparisons are made with the student's t-test. However, it has not been previously shown that the differences have a normal distribution or any method of normalization. It is not advisable to take it for granted, given that the measurements are subjective, not psychophysiological, responses that present normality. It is suggested to review whether all multiple comparisons can be analyzed with the student's t-test or non-parametric comparisons (natural in discrete measures) should be applied. In addition, it is also recommended to enter the effect size of such differences (d or r, depending on whether it is parametric or non-parametric). Also, in line 341 the authors discuss an idea (what they attribute to the data found). This should go in the discussion section.
Comments on the Quality of English LanguageThe English language requires a minor technical review of the entire document. In the abstract alone, 5 inaccuracies were found that must be improved.
Author Response
We gratefully thanks for the precious time and effort that you spent on reviewing our manuscript and making constructive remarks. We have responded to your comments point by point, please see the attachment.

Reviewer 5 Report
Comments and Suggestions for Authors
Please diversify the use of "additional", "also", "suggest".
Please observe the attached file for some more improvement suggestions.

Comments on the Quality of English LanguageMinor revisions are necessary.
Author Response
We would like to express our sincere thanks to you for your valuable suggestions and comments. And thank you for taking time out of your busy schedule to review our manuscript. We have responded to your comments point by point, please see the attachment.

Round 2
Reviewer 4 Report
Comments and Suggestions for Authors
Las modificaciones son satisfactorias.